# HP-YOLOv8: High-Precision Small Object Detection Algorithm for Remote Sensing Images

**DOI:** 10.3390/s24154858

**Published:** 2024-07-26

**Authors:** Guangzhen Yao, Sandong Zhu, Long Zhang, Miao Qi

**Affiliations:** School of Information Science and Technology, Northeast Normal University, Changchun 130117, China; yaoguangzhen@nenu.edu.cn (G.Y.); zhusandong@nenu.edu.cn (S.Z.); longzhang@nenu.edu.cn (L.Z.)

**Keywords:** YOLOv8, small object detection, remote sensing images, attention mechanism, feature fusion

## Abstract

YOLOv8, as an efficient object detection method, can swiftly and precisely identify objects within images. However, traditional algorithms encounter difficulties when detecting small objects in remote sensing images, such as missing information, background noise, and interactions among multiple objects in complex scenes, which may affect performance. To tackle these challenges, we propose an enhanced algorithm optimized for detecting small objects in remote sensing images, named HP-YOLOv8. Firstly, we design the C2f-D-Mixer (C2f-DM) module as a replacement for the original C2f module. This module integrates both local and global information, significantly improving the ability to detect features of small objects. Secondly, we introduce a feature fusion technique based on attention mechanisms, named Bi-Level Routing Attention in Gated Feature Pyramid Network (BGFPN). This technique utilizes an efficient feature aggregation network and reparameterization technology to optimize information interaction between different scale feature maps, and through the Bi-Level Routing Attention (BRA) mechanism, it effectively captures critical feature information of small objects. Finally, we propose the Shape Mean Perpendicular Distance Intersection over Union (SMPDIoU) loss function. The method comprehensively considers the shape and size of detection boxes, enhances the model’s focus on the attributes of detection boxes, and provides a more accurate bounding box regression loss calculation method. To demonstrate our approach’s efficacy, we conducted comprehensive experiments across the RSOD, NWPU VHR-10, and VisDrone2019 datasets. The experimental results show that the HP-YOLOv8 achieves 95.11%, 93.05%, and 53.49% in the mAP@0.5 metric, and 72.03%, 65.37%, and 38.91% in the more stringent mAP@0.5:0.95 metric, respectively.

## 1. Introduction

In recent years, with the quantity and quality of remote sensing images having significantly improved, object detection has become crucial for the automated analysis of these images. This technology not only enables rapid and accurate object classification and tracking but also finds extensive applications in civil, commercial, and military domains, such as in drones [1,2,3,4,5], intelligent traffic monitoring [6,7,8,9,10], and aerospace [11,12,13,14]. Currently, within the domain of deep learning, object detection technologies are primarily categorized into two primary categories: region proposal-driven methods, exemplified by R-CNN and its variants, which operate by identifying potential areas of interest for detection; and regression-based strategies, such as the SSD and YOLO series, known for their fast processing speeds and suitability for real-time detection tasks.

Although object detection technologies have seen considerable progress, several challenges still persist. These challenges include complex backgrounds, low image quality, high diversity in object arrangement, and arbitrary object orientations. Especially in images with complex backgrounds, low resolution, and densely distributed objects, the detection of small objects often yields unsatisfactory results.

In response to these issues, researchers have suggested numerous enhancements. For example, scaling and merging feature maps successfully preserved small object information while enhancing feature expressiveness [15]. By combining the deep semantic information with the shallow localization information, the effectiveness of feature fusion was significantly improved [16]. Additionally, existing studies had explored the integration of attention mechanisms, feature embedding, and feature transmission techniques to optimize the feature fusion process [17,18,19]. Significant progress was achieved by applying feature pyramid networks (FPNs) [20] and transformer models equipped with self-attention mechanisms [21]. Due to their precision and scalability [22], the YOLO algorithm suite had been extensively applied in object detection [23]. By improving the backbone network structure [24,25,26,27,28,29], it was possible to effectively extract attributes from objects with intricate shapes and appearances. The refined feature fusion method [25,30] also contributed to achieving superior detection results. Additionally, adjustments to the bounding box regression loss function [31,32,33,34] substantially enhanced the overall performance of the network. Drawing on these advanced technologies, we propose an improved algorithm named HP-YOLOv8, integrating three core technologies: the C2f-D-Mixer (C2f-DM) module, Bi-Level Routing Attention in Gated Feature Pyramid Network (BGFPN) feature fusion technique, and Shape Mean Perpendicular Distance Intersection over Union (SMPDIoU) loss function, focused on enhancing the detection precision of small objects within remote sensing images.

Our main contributions are as follows:We design and implement the C2f-DM module as a replacement for the current C2f module. The module efficiently integrates local and global information, significantly improving the ability to capture features of small objects while effectively mitigating detection precision issues caused by object overlap.We propose a feature fusion technique based on the attention mechanism, named BGFPN. This technique utilizes an efficient feature aggregation network and re-parameterization technology to optimize the interaction of information between feature maps of different scales. Through the Bi-level Routing Attention (BRA) mechanism, it effectively captures key feature information of small objects.We propose a SMPDIoU loss function. This approach thoroughly accounts for the shape and dimensions of the detection boxes, strengthens the model’s focus on the attributes of detection boxes, and provides a more accurate bounding box regression loss calculation method.

## 2. Related Work

Within the realm of remote sensing object detection, key technologies primarily encompass feature extraction, feature fusion, and the optimization of bounding box regression loss functions. Feature extraction aims to derive key information from remote sensing images to facilitate accurate object identification. Feature fusion enhances the model’s recognition and classification capabilities by integrating features from different levels. Furthermore, the bounding box regression loss function is essential for accurately predicting the position and dimensions of objects. Together, these technologies improve the precision and effectiveness of remote sensing object recognition algorithms.

### 2.1. Feature Extraction

The architecture of backbone networks is designed to efficiently capture and combine information from multiple scales. For instance, YOLOv2 [35] significantly improved its feature extraction capabilities by substituting its original network with DarkNet-19, which enhanced both detection precision and speed. YOLOv3 [36] implemented a multi-scale prediction approach similar to FPN and introduced a more robust backbone network, DarkNet-53, further boosting the efficiency and precision of feature extraction. YOLOv5 integrated Focus and Cross Stage Partial (CSP) modules into its backbone network, enhancing detection speed and precision through advanced training optimization strategies. LSKNet [37], a lightweight large selective kernel network, dynamically adjusted the spatial receptive field to better accommodate diverse objects in remote sensing images. YOLOv8 replaced the earlier C3 module with the C2f module, maintaining the network’s lightweight structure while facilitating a richer flow of gradient information. Specifically targeting micro unmanned aerial vehicle detection, YOLO-Drone [24] incorporated the SPD-Conv module. In the identification of small objects within remote sensing imagery, LAR-YOLOv8 [25] replaced the C2f modules in the YOLOv8 backbone with DCN-C2f modules. In further research based on the YOLOv8 model, we introduce the C2f-DM module to replace the C2f module before the detection head, significantly enhancing the capability to capture features of small objects and effectively mitigating issues of detection accuracy caused by object overlaps.

### 2.2. Feature Fusion

FPN effectively integrates features of different resolutions from the backbone network, achieving multi-scale feature fusion and significantly enhancing network performance and robustness. Given the complex backgrounds and noise inherent in remote sensing images, implementing an attention mechanism can more effectively isolate critical features of small objects. By dynamically learning the distribution or weights of attention, this mechanism enabled the model to adaptively concentrate on essential information as dictated by the context and task requirements, thus greatly enhancing the precision of object detection.

Methods for feature fusion and attention mechanisms have progressed from focusing solely on either spatial or channel attention to incorporating a mix of both channel and spatial attentions, along with the adoption of self-attention mechanisms. For example, TPH-YOLOv5 [38] incorporated the Convolutional Block Attention Module (CBAM), successfully pinpointing attention regions in densely populated scenes. Similarly, YOLO-Drone [24] improved drone detection performance by integrating the Global Attention Mechanism (GAM) into the neck section. Moreover, the SuperYOLO [39] fused multimodal data and super-resolution technology, enhancing the ability to detect small objects. LAR-YOLOv8 [25] designed a bi-directional feature pyramid network that effectively extracted shallow features using the attention mechanism and optimized feature fusion. Ref. [40] proposed a Multi-Information Perception Attention Module (MIPAM) that significantly enhanced the precision and real-time performance of underwater object detection by gathering multi-dimensional information, interacting across dimensions, and activating attention mechanisms. This approach effectively addressed the challenges posed by poor imaging quality, harsh underwater environments, and the concealed nature of objects. Article [41] introduced a method called Manifold-based Incomplete Multi-view Clustering via Bi-consistency Guidance (MIMB). This method constructed a bi-consistency guided consistency representation learning framework and combined manifold embedding with reverse projection regularization to explore and balance the structural information between recovered data and original data, thereby significantly improving the accuracy and robustness of clustering. Ref. [42] developed a method known as Graph-Collaborated Auto-Encoder Hashing for Multi-view Binary Clustering (GCAE), which, by integrating affinity graphs with auto-encoders, effectively mined and integrated compatible and complementary information from different views to address the binary clustering challenges in multi-view data. In this paper, we introduce a feature fusion method named BGFPN that utilizes attention mechanisms. This approach leverages an efficient feature aggregation network and reparametrization techniques to enhance the exchange of information across feature maps of varying scales. Additionally, it adeptly captures essential feature information of small objects using the BRA mechanism.

### 2.3. Optimization of Bounding Box Regression Loss Function

In tasks related to object detection, the loss function for bounding box regression is crucial. Previously, a widely utilized method was Intersection over Union (IoU) [43], which evaluates overlap by measuring the ratio of the shared area to the combined area of the predicted and true boxes. However, IoU exhibited clear limitations when addressing small objects, particularly when the predicted and true boxes had no intersection, yielding an IoU value of zero, which could lead to gradient vanishing and hinder the model from effectively learning the features of small objects. To address these issues, researchers proposed several improvements, such as Generalized Intersection over Union (GIoU) [44], Distance Intersection over Union (DIoU) [45], and Complete Intersection over Union (CIoU). GIoU introduced the concept of the smallest enclosing box to compute a non-zero loss value, solving the gradient vanishing issue when no overlap existed between the predicted and further boxes. DIoU and CIoU actual considered the differences in aspect ratio and center distance to enhance regression precision. Additionally, Cascade R-CNN [31] consisted of a series of detectors that used progressively increasing IoU thresholds during training to more stringently exclude neighboring false positives. CenterNet [32] adopted a corner-based keypoint detection approach, thereby circumventing the traditional requirement for anchor boxes. While these methods made progress in handling small objects and bounding boxes with extreme aspect ratios, they primarily focused on the geometric relationships between bounding boxes and overlooked the potential impact of the bounding boxes’ shapes and scales on regression effectiveness. To enhance small object detection further, we introduce a novel approach: SMPDIoU. This method combines the advantages of Shape Intersection over Union (SioU) [46] and Mean Perpendicular Distance Intersection over Union (MPDIoU) [47], comprehensively considering the shape and proportion of bounding boxes, effectively compensating for the shortcomings of IoU and its derivative methods.

## 3. Fundamentals of the YOLO v8 Model

Since the YOLO model was first introduced, the series has undergone multiple updates and iterations, with continually enhanced performance. As the most recent development in the YOLO model series, YOLOv8 represents the current pinnacle of technology. Its architecture, depicted in Figure 1, comprises three main components: the backbone, the detection head, and the neck.

Backbone: This section mainly conducts feature extraction using a sequence of C2f modules, Conv modules, and SPPF modules. YOLOv8 introduces a new C2f module to replace the previous C3 module, which, while maintaining a lightweight structure, promotes richer gradient information flow.

Detection head: YOLOv8 has added a decoupled head module, an innovative design that separates classification and localization tasks, effectively mitigating potential conflicts between these two tasks and, thereby, enhancing the overall efficacy of the model.

Neck: YOLOv8 follows the Path Aggregation-Feature Pyramid Network (PA-FPN) design philosophy but simplifies the convolution process in the upsampling stage to enhance performance.

Overall, YOLOv8 not only inherits the efficiency of the YOLO series but also innovates in model structure and algorithm, making it perform exceptionally well in handling complex detection tasks.

## 4. Methodology

### 4.1. Framework Overview 

YOLOv8 demonstrates outstanding performance across multiple application domains. However, in remote sensing object detection, challenges persist in accurately detecting small objects. These challenges manifest primarily in two aspects: First, when neural networks extract features from images, features of small objects may be obscured by larger surrounding objects, causing a loss of critical information. This can result in small objects being overlooked during the learning phase, thus impacting the precision of detection. Second, in complex scenes with multiple object interactions, small objects are more susceptible to false positives and omissions. Compared to larger objects, they are more likely to be obscured or to overlap with other objects, making visual distinction and localization more difficult. To tackle these challenges, we introduce HP-YOLOv8, an improved version of the YOLOv8 algorithm specifically designed for detecting small objects in remote sensing (as depicted in Figure 2).

As shown in Figure 3. Firstly, we designed a continuously stacking and fusing module named C2f-DM (detailed in Section 4.2). The C2f-DM module, by integrating local and global information, enhances the capability to capture features of small objects and effectively alleviates the detection accuracy problems caused by object overlaps.

Secondly, we introduced an attention-based feature fusion technique, named BGFPN (detailed in Section 4.3). This technique utilizes an efficient feature aggregation network and reparameterization technology to optimize the interaction of information between feature maps at various scales. Additionally, by introducing the BRA mechanism, BGFPN can more effectively capture critical feature information of small objects.

Lastly, we introduced a novel IoU loss function calculation method named SMPDIoU (detailed in Section 4.4). This method comprehensively considers the shape and size of detection boxes, thereby strengthening the model’s focus on the attributes of detection boxes. It not only adjusts the shape and position of bounding boxes more accurately but also adapts the regression strategy according to the varying sizes of objects. Moreover, SMPDIoU, by considering the perpendicular distance between two target boxes, provides a more precise bounding box regression loss calculation method.

### 4.2. C2f-DM Module 

The YOLOv8 backbone network mainly consists of stacks of simple convolutional modules. This design can cause small object features to be overshadowed by those of larger surrounding objects during image extraction, leading to the loss of crucial information. To improve the network’s capability to process small objects, we introduced a novel module called C2f-DM, which replaces the existing C2f module before the detection head.

As shown in Figure 4, in the C2f module, we use the DM-bottleneck structure embedded with the Dual Dynamic Token Mixer (D-Mixer) [48] to replace the original bottleneck. This configuration merges the benefits of both convolution and self-attention mechanisms while introducing a robust inductive bias for handling uniformly segmented feature segments. It achieves the dynamic integration of local and global information, considerably extending the network’s effective field of view. The module processes the input feature map in two segments: one via Input-dependent Depth-wise Convolution (IDConv) and the other through Overlapping Spatial Reduction Attention (OSRA). Subsequently, the outputs of these two parts are merged.

Specifically, we consider a feature map *X* of dimensions $R^{C\times H\times W}$. This map is initially split into two sub-maps, $\left\{X_{1},X_{2}\right\}$, along the channel dimension, each with dimensions $R^{C/2\times H\times W}$. Subsequently, $X_{1}$ is processed by the OSRA, while $X_{2}$ is handled by IDConv, resulting in new feature maps $\left\{X_{1}^{\prime },X_{2}^{\prime }\right\}$ of the same dimensions. These maps are subsequently combined along the channel dimension, resulting in the final output feature map $X^{\prime }$ with dimensions $R^{C\times H\times W}$. Finally, the Compression Token Enhancer (STE) enables efficient local token aggregation. The D-Mixer performs the following sequence of operations:(1)$$\begin{array}{cc} \; & X_{1},X_{2}=\mathrm{Split}\left(X\right)\\ \; & X^{\prime }=\mathrm{Concat}\left(\mathrm{OSRA}\left(X_{1}\right),\mathrm{IDConv}\left(X_{2}\right)\right)\\ \; & Y=\mathrm{STE}(X^{\prime }) \end{array}$$

In the IDConv module, the input feature map *X* with dimensions $R^{C\times H\times W}$ initially undergoes adaptive average pooling to gather spatial context and reduce spatial dimensions to $K^{2}$, ensuring the capture of global information. Following this, the map is passed through two consecutive 1×1 convolution layers to create an attention map $A^{\prime }$ of dimensions $R^{\left(G\times C\right)\times K^{2}}$, enabling the network to dynamically focus on important regions of the input feature map, thereby achieving local information integration. where *G* represents the quantity of attention groups. The map $A^{\prime }$ is reshaped to $R^{G\times C\times K^{2}}$ and a softmax function is applied across the *G* dimension to produce the attention weights *A* in $R^{G\times C\times K^{2}}$. These weights *A* are then multiplied element-wise with a set of learnable parameters *P*, also in $R^{G\times C\times K^{2}}$, and aggregated over the *G* dimension to form the tailored deep convolution kernel *W* in $R^{C\times K^{2}}$. This process dynamically adjusts the convolutional kernel weights based on the different characteristics of the input feature map, integrating global and local information. The entire process of IDConv can be expressed as
(2)$$\begin{array}{cc} \; & A^{\prime }={\mathrm{Conv}}_{1\times 1}^{C/r\to (G\times C)}\left({\mathrm{Conv}}_{1\times 1}^{C\to C/r}\left(\mathrm{AdaptivePool}\left(X\right)\right)\right)\\ \; & A=\mathrm{Softmax}(\mathrm{Reshape}\left(A^{\prime }\right))\\ \; & W=\sum_{i=0}^{G}P_{i}\cdot A_{i} \end{array}$$

In the OSRA module, a technique known as Overlapping Space Reduction (OSR) is employed to improve the spatial structure representation within the self-attention mechanism. This technique employs larger and overlapping patches to more effectively capture spatial information near patch boundaries, thus improving the representation of spatial structures in the self-attention mechanism. This process not only ensures the capture of local information and the expression of features but also achieves the integration of global information through the overlapping parts of the patches. The entire process of OSRA can be expressed as
(3)$$\begin{array}{cc} \; & Y=\mathrm{OSR}(X)\\ \; & Q=\mathrm{Linear}(X)\\ \; & K,V=\mathrm{Split}(\mathrm{Linear}(Y+\mathrm{LR}(Y)))\\ \; & Z=\mathrm{Softmax}\left(\frac{QK^{T}}{\sqrt{d}+B}\right)V \end{array}$$
where *B* denotes the relative position bias matrix, *d* represents the number of channels per attention head, and LR(·) refers to the Local Refinement Module, implemented using a 3×3 depth-wise convolution.

In the STE module, a 3×3 depth-wise convolution enhances local relationships, 1×1 convolutions for channel squeezing and expansion reduce computational cost, and a residual connection ensures representational capacity. This setup integrates both local and global information. STE can be represented as
(4)$$\begin{array}{c} \mathrm{STE}\left(X\right)={\mathrm{Conv}}_{1\times 1}^{C/r\to C}\left({\mathrm{Conv}}_{1\times 1}^{C\to C/r}\left({\mathrm{DWConv}}_{3\times 3}\left(X\right)\right)\right)+X \end{array}$$

By integrating the C2f-DM module, we demonstrate significant advantages in processing complex remote sensing imagery and small object detection compared to traditional frameworks like YOLOv8 and other detection algorithms. Currently, many detection algorithms primarily rely on simple stacks of convolutional modules. While this approach offers computational efficiency, it may lack the required flexibility and precision when dealing with complex scenes or small objects. In contrast, the C2f-DM module dynamically integrates local and global information, enabling more precise feature extraction in scenarios involving small objects and complex backgrounds. This capability is particularly crucial for remote sensing imagery applications that require extensive fields of view and meticulous feature analysis. The OSRA technology within the C2f-DM module significantly improves the capture of spatial features through overlapping space reduction techniques, especially around object edges. Simultaneously, IDConv dynamically adjusts the convolution kernels based on input, enhancing the module’s sensitivity to small objects and effectively reducing information loss. These characteristics allow the C2f-DM module to surpass current detection methods in providing more efficient and precise detection performance.

Although the C2f-DM introduces relatively complex mechanisms, it optimizes the use of computational resources through STE technology by performing channel squeezing and expansion operations after deep convolution. This ensures that the module remains highly efficient even in resource-constrained environments. Compared to traditional methods, the design of the C2f-DM module allows for more flexible network structure adjustments to accommodate different application needs. By tuning parameters within STE, an optimal balance between precision and speed can be found, tailored to specific tasks without the need for redesigning the entire network architecture.

Furthermore, the design of the C2f-DM module incorporates dynamic adjustment capabilities, enabling it to flexibly handle input features of varying scales and complexities and automatically adjust processing strategies to suit different inputs and scene conditions. This trait is particularly key for remote sensing image analysis, as these images often involve extensive geographic and environmental variations, along with constantly changing lighting and weather conditions. Therefore, the high adaptability of the C2f-DM module allows it to excel in scenarios with complex backgrounds or multi-scale objects, showcasing exceptional optimization potential and robust adaptability. Compared to existing methods, the adaptive capability of the C2f-DM is more pronounced, reducing reliance on manual intervention and significantly enhancing usability and flexibility, especially under a wide range of practical application conditions.

### 4.3. Bi-Level Routing Attention in Gated Feature Pyramid Network

#### 4.3.1. Improved Feature Fusion Method

FPNs achieve multi-scale feature fusion by aggregating different resolution features from the backbone network. This approach not only boosts network performance but also improves its robustness, and has been proven to be extremely crucial and effective in object detection. Nonetheless, the current YOLOv8 model only adopts the PANet structure. This approach can be easily disrupted by normal-sized objects when processing small-sized objects, potentially leading to a gradual reduction or even complete disappearance of small object information. Additionally, there are issues with the precision of object localization in this model. To tackle these challenges, we propose BGFPN, a new feature fusion method.

We incorporated a top-down pathway to transmit high-level semantic feature information, guiding subsequent network modules in feature fusion and generating features with enhanced discriminative capacity. Additionally, a BRA [49] mechanism was introduced to extract information from very small target layers (as shown in Figure 5). This is a structure that uses sparse operations to efficiently bypass the most irrelevant areas, creating powerful discriminative object features.

BGFPN innovates on the basis of the Re-parameterized Gated Feature Pyramid Network (RepGFPN) [50] through an efficient feature aggregation network and reparameterization techniques, optimizing the information interaction between different scale feature maps. This architecture improves the model’s handling of multi-scale information and efficiently merges spatial details with low-level and high-level semantic information. Although a large number of upsampling and downsampling operations was introduced to enhance interactions between features, a method was adopted to remove additional upsampling operations that cause significant latency, improving real-time detection speed.

When dealing with feature fusion issues between different scales, the model eliminates traditional 3 × 3 convolution modules and introduces Cross Stage Partial Stage (CSPStage) [51] modules with a reparameterization mechanism. This module uses an efficient layer aggregation network connection as a feature fusion block, utilizing Concat operations to connect inputs from different layers. This allows the model to integrate shallow and deep feature maps, thereby obtaining rich semantic and positional information and high pixel points, enhancing the receptive field and improving model precision. RepConv [52], as a representative of the reparameterized convolution module, achieves branch fusion during inference, which not only reduces inference time but also increases inference speed.

Furthermore, to more precisely address the detection of small targets, we introduced dilated convolution technology [53]. This technology enhances feature extraction capabilities by expanding the convolution kernel’s receptive field without adding extra computational burden. This approach bypasses pooling operations, thus maintaining the high resolution of the feature maps. This is critical for the precise localization and identification of small objects within images, greatly enhancing the model’s detection precision in intricate scenes, particularly those with visual noise.

#### 4.3.2. Bi-Level Routing Attention

In remote sensing images, complex backgrounds and severe noise often obscure small objects. Incorporating an attention mechanism into the network greatly enhances the capture of essential feature information, thus enhancing object detection precision. However, traditional attention mechanisms impose a considerable computational load when dealing with extremely small object layers, especially at high resolutions. To mitigate this, we have integrated a BRA mechanism tailored for vision transformers into the neck structure of YOLOv8. As shown in Figure 6, this mechanism first filters out irrelevant large-area features at a coarser region level, then focuses at a finer token level, and dynamically selects the most pertinent key–value pairs for each query. This strategy not only saves computational and memory resources but also greatly improves the precision of detecting small objects.

Initially, we divided a two-dimensional input feature map $X\in R^{H\times W\times C}$ into S×S non-overlapping regions, each containing $\frac{HW}{S^{2}}$ feature vectors. This reshaped *X* into $X^{r}\in R^{S^{2}\times \frac{HW}{S^{2}}\times O}$. Linear projections then generated the queries, keys, and values tensors $Q,K,V\in R^{S^{2}\times \frac{HW}{S^{2}}\times C}$. We proceeded to construct a directed graph that maps the attention relations between these regions. We averaged the regions within *Q* and *K* to create region-level queries and keys $Q^{r},K^{r}\in R^{S^{2}\times C}$. The adjacency matrix for the region-to-region affinity graph was subsequently computed by multiplying the transpose of $Q^{r}$ with $K^{r}$.
(5)$$\begin{array}{c} A^{r}=Q^{r}{\left(K^{r}\right)}^{T} \end{array}$$

From here, we identified the top-k regions with the highest similarity for each region in the adjacency matrix through row-wise operations. These indices were then annotated in the region-to-region routing index matrix, where top-k indicates the number of regions of interest within BGFPN.
(6)$$\begin{array}{c} I^{r}=topkIndex\left(A^{r}\right) \end{array}$$

Utilizing the inter-region routing index matrix $I^{r}$, we then implemented fine-grained token-to-token attention. Initially, we gathered the key and value tensors, denoted as $V^{g}=\mathrm{gather}\left(V,I^{r}\right)$ and $K^{g}=\mathrm{gather}\left(K,I^{r}\right)$. Following the integration of Local Context Enhancement (LCE), attention was directed towards these gathered key–value pairs to generate the output:(7)$$\begin{array}{c} O=\mathrm{Attention}\left(Q,K^{g},V^{g}\right)+\mathrm{LCE}\left(V\right) \end{array}$$

Depicted in Figure 2, in the BGFPN structure, we incorporate the BRA mechanism after each C2f module during the upsampling process, before downsampling, and before feature fusion. By adding the BRA module before the upsampling step, the features can be focused on earlier, allowing for a more precise handling of small object information, significantly enhancing the object’s recognition and localization performance. Moreover, by introducing the BRA module after each C2f module during the downsampling process, it ensures that even after feature simplification, the model can still sensitively capture details, strengthening the recognition of key information. Especially by introducing the BRA module before feature fusion, this can screen key areas at the macro level and conduct in-depth detail attention at the token level, ensuring that the network prioritizes key information in the image before integrating features, further improving the detection precision of small objects. This integrated attention mechanism effectively isolates crucial information in intricate settings while amplifying focus on fundamental features, thereby markedly boosting the precision of detecting small objects.

By integrating the BGFPN feature fusion technology, we have effectively optimized the information interaction between differently scaled feature maps. This design not only enhances the model’s ability to process multi-scale information, but also effectively merges high-level semantic information with low-level spatial information. Compared to other FPNs, BGFPN more meticulously handles features from details to the whole, thus providing richer and more accurate information in complex scenarios. Unlike traditional methods, which often struggle with small object detection, especially in noisy or complex backgrounds, BGFPN demonstrates significant advantages by incorporating dilated convolution technology. This not only expands the receptive field of the convolution kernels to enhance feature extraction capabilities but also avoids additional computational burdens. This strategy maintains a high spatial resolution of the feature maps, significantly enhancing the model’s precision in detecting small objects in complex scenes.

Furthermore, unlike conventional methods that typically incorporate attention mechanisms only at the backbone and neck connections of networks, our model introduces the BRA mechanism after each C2f module during both upsampling and downsampling processes, as well as before feature fusion. This approach not only captures key features of small objects more effectively but also optimizes small object detection and localization, significantly improving overall detection precision.

Although BGFPN employs numerous upsampling and downsampling operations to enhance interaction between features, it eliminates additional upsampling operations that cause significant latency issues, effectively improving the model’s speed in real-time detection. This improvement is crucial for scenarios requiring rapid responses, such as video surveillance or autonomous driving, as it significantly reduces processing time while ensuring efficient feature handling.

In summary, BGFPN not only improves the precision and speed of detection but also exhibits stronger adaptability and performance in handling complex and variable scenes, particularly surpassing many existing frameworks in terms of small object detection requirements.

### 4.4. Shape Mean Perpendicular Distance Intersection over Union

The bounding box regression loss function is crucial in object detection tasks. Researchers consistently propose various improved methods, such as GIoU [44], DIoU [45], and CIoU. While these approaches have enhanced the handling of small objects and bounding boxes with extreme aspect ratios, they still mainly emphasize the geometric relationship between bounding boxes, overlooking the influence of the bounding box’s own shape and scale on regression results.

To enhance small object detection, we introduced a new method called SMPDIoU. This method combines the advantages of SioU [46] and MPDIoU [47], comprehensively considering the shape and scale of the bounding boxes, thus addressing the deficiencies of IoU and its improved versions.Furthermore, SMPDIoU incorporates a detailed regression loss calculation method centered on the vertical distance between two bounding boxes. This approach not only markedly enhances the precision of detecting large objects but also excels in detecting small objects, efficiently addressing prevalent issues in small object detection. The specific calculation formula is provided below:(8)$$\begin{array}{c} \mathrm{SMPDIoU}=\alpha \cdot \left(\mathrm{IoU}-\frac{\Delta +\Omega }{2}+\left(1-\alpha \right)\right)\cdot \left(\frac{\mathrm{IoU}+(1-D_{perpendicular\_norm})}{2}\right) \end{array}$$
where α is a weight parameter, used to balance the influences of SIOU and MPDIoU, which can be adjusted according to specific application scenarios. In this model, distance loss (Δ) and shape loss (Ω) play a key role. By measuring the spatial distance and shape discrepancies between the actual and predicted boxes, SMPDIoU effectively reduces the angular differences between the anchor and true boxes in the horizontal or vertical directions, thus accelerating the convergence process of bounding box regression. The distance loss (Δ) is defined by the following equation:(9)$$\begin{array}{c} \Delta =\sum_{t=w,h}\left(1-e^{-\gamma \rho_{t}}\right) \end{array}$$
where γ=2−Λ, and $\rho_{t}$ is the standardized distance between the centers of the true and predicted bounding boxes, calculated as follows:(10)$$\begin{array}{c} \rho_{x}={\left(\frac{x_{c}^{pre}-x_{c}^{gt}}{w_{c}}\right)}^{2},\rho_{y}={\left(\frac{y_{c}^{pre}-y_{c}^{gt}}{h_{c}}\right)}^{2} \end{array}$$
As shown in Figure 7, $x_{c}^{pre}$ and $y_{c}^{pre}$ are the center coordinates of the predicted bounding box, while $x_{c}^{gt}$ and $y_{c}^{gt}$ are the center coordinates of the true bounding box. Additionally, $h_{c}^{pre}$, $w_{c}^{pre}$, $h_{c}^{gt}$, and $w_{c}^{gt}$ denote the respective heights and widths of the predicted and actual bounding boxes. The coefficient γ related to the angle is calculated by the following equation:(11)$$\begin{array}{c} \Lambda =1-2*\sin^{2}\left(\arcsin\left(\frac{h_{c}}{\sigma }\right)-\frac{\pi }{4}\right)=\cos\left(2*\left(\arcsin\left(\frac{h_{c}}{\sigma }\right)-\frac{\pi }{4}\right)\right) \end{array}$$
where σ denotes the Euclidean distance from the center of the predicted bounding box to the center of the true bounding box, calculated as follows:(12)$$\begin{array}{c} \sigma =\sqrt{{\left(x_{c}^{gt}-x_{c}^{pre}\right)}^{2}+{\left(y_{c}^{gt}-y_{c}^{pre}\right)}^{2}} \end{array}$$
where $h_{c}$ represents the discrepancy in the y-axis distances between the minimum and maximum extents of the true and predicted bounding boxes, expressed as:(13)$$\begin{array}{c} h_{c}=\max\left(y_{c}^{gt},y_{c}^{pre}\right)-\min\left(y_{c}^{gt},y_{c}^{pre}\right) \end{array}$$

The equation for shape loss (Ω) is given below:(14)$$\begin{array}{c} \Omega =\sum_{t=w,h}\left(1-e^{-\omega_{t}}\right)\theta \end{array}$$
where θ=4 and $\omega_{t}$ represents the proportional variances in height and width across the bounding boxes, calculated in the following manner:(15)$$\begin{array}{c} \omega_{w}=\left|\frac{w_{c}^{pre}-w_{c}^{gt}}{\max(w_{c}^{pred},w_{c}^{gt})}\right|,\omega_{h}=\left|\frac{h_{c}^{pre}-h_{c}^{gt}}{\max(h_{c}^{pre},h_{c}^{gt})}\right| \end{array}$$

In order to enhance the precision of assessing the spatial alignment between true and predicted bounding boxes, the model integrates the computation of the vertical distance separating their centers:(16)$$\begin{array}{c} \mathrm{SMP}=\frac{\mathrm{IoU}+(1-D_{perpendicular\_norm})}{2} \end{array}$$
where $D_{perpendicular\_norm}$ is the normalized vertical distance, which varies between 0 and 1. This normalized distance is derived from the Euclidean distance $D_{perpendicular}$ between the true and predicted bounding box centers, relative to the maximum distance $D_{\max}$ that serves as the normalization reference:(17)$$\begin{array}{c} D_{perpendicular\_norm}=\frac{D_{perpendicular}}{D_{\max}}\\ D_{perpendicular}=\sqrt{{\left(x_{c}^{pre}-x_{c}^{gt}\right)}^{2}+{\left(y_{c}^{pre}-y_{c}^{gt}\right)}^{2}} \end{array}$$
(18)$$\begin{array}{c} x_{c}^{pre}=\frac{\left(x_{1}^{pre}+x_{2}^{pre}\right)}{2},y_{c}^{pre}=\frac{\left(y_{1}^{pre}+y_{2}^{pre}\right)}{2}\\ x_{c}^{gt}=\frac{\left(x_{1}^{gt}+x_{2}^{gt}\right)}{2},y_{c}^{gt}=\frac{\left(y_{1}^{gt}+y_{2}^{gt}\right)}{2} \end{array}$$

## 5. Experiments

### 5.1. Experimental Setup 

Experimental Environment. All experiments described in this paper were conducted using a defined system setup to ensure effective implementation and the reproducibility of outcomes. For detailed specifics, refer to Table 1.

Hyperparameter Settings. During our experimental procedures, throughout the training phase, we applied a learning rate decay method. We performed the training over 200 epochs to maintain stability, during which the learning rate progressively reduced. Additionally, other configurations adhered to the default settings of the original YOLOv8. The specifics of the training hyperparameters can be found in Table 2.

Datasets. In the field of remote sensing image analysis, selecting datasets with extensive coverage and diverse scenes is crucial. We carefully chose three datasets—RSOD [54], NWPU VHR-10 [55], and VisDrone2019 [56]—that have unique characteristics and advantages to ensure a comprehensive representation of potential scenarios in remote sensing imagery.

The RSOD dataset focuses on infrastructure-related objects, including aircrafts, playgrounds, overpasses, and oil tanks. These are key objects in remote sensing image analysis. The dataset consists of 446 images containing 4993 aircraft, 189 images with 191 playgrounds, 176 images with 180 overpasses, and 165 images with 1586 oil tanks. These categories are widely distributed across urban and rural environments, providing a rich source of training material for models. The detection of aircraft and oil tanks is particularly critical for military and industrial applications. We randomly divided the RSOD dataset into training, validation, and testing sets in an 8:1:1 ratio.

The NWPU VHR-10 dataset offers high-resolution satellite images from various global locations, primarily sourced from Google Earth and the Vaihingen database. This dataset includes 800 images with a resolution of 1000 × 1000 pixels, 715 of which are color images from Google Earth with spatial resolutions ranging from 0.5 to 2 m; the remaining 85 are color infrared images sharpened from 0.08 to one meter resolution from Vaihingen data. With a total of 3775 object instances, this dataset showcases a range of geographical spatial objects from urban buildings to natural landscapes, including airplanes, baseball diamonds, basketball courts, bridges, harbors, ground track fields, ships, storage tanks, tennis courts, and vehicles. Its wide geographic coverage and diverse environmental conditions enable the effective assessment of model performance in various terrains and object detections. This dataset was also divided into training, validation, and testing sets in an 8:1:1 ratio.

The VisDrone2019 dataset, developed jointly by Tianjin University and the AISKYEYE data mining team, includes 288 video segments, 10,209 static images, and 261,908 video frames. These were captured by various types of drone cameras across different scenes, weather conditions, and lighting settings. Covering a broad area across 14 different cities in China, including urban and rural settings, this dataset encompasses a wide variety of objects such as pedestrians, cars, bicycles, and awning tricycles, with scene densities ranging from sparse to crowded. All images have been manually annotated, documenting over 2.6 million object bounding boxes, and detailing key attributes such as scene visibility, object categories, and occlusion. From VisDrone2019, we randomly selected 2000 images for the training set, and allocated 100 images each to the validation and testing sets.

The collective use of these three datasets not only spanned a wide range of scenarios from infrastructure to natural landscapes and from static to dynamic objects, but also covered different scene densities from high to sparse. This comprehensive data coverage ensures that our research in remote sensing image analysis is highly applicable and reliable across various applications.

Evaluation Metrics. To assess the proposed HP-YOLOv8 model, we utilized three commonly employed metrics in object detection: recall(R), precision(P), F1 score and Mean Average Precision (mAP). The formulas are as follows:(19)$$\begin{array}{c} P=\frac{TP}{TP+FP},\;R=\frac{TP}{TP+FN},\;F_{1}=2\times \frac{P\times R}{P+R}\\ AP=\int_{0}^{1}P\left(R\right)\;dR,\;mAP=\frac{1}{N}\sum_{i=1}^{N}AP_{i} \end{array}$$
where *N* denotes the total number of classes. The mAP metric assesses the mean AP across all classes. mAP@0.5 represents the average precision at an IoU threshold of 0.5, while mAP@0.5:0.95 computes the average precision over IoU thresholds from 0.5 to 0.95, incrementally increasing by 0.05. This metric provides a more comprehensive performance evaluation, covering a spectrum from looser to stricter matching criteria.

### 5.2. Overall Performance of HP-YOLOv8

Classification Evaluation. We extensively experimented with the RSOD dataset to evaluate how the performance of the traditional YOLOv8 model compared to our newly developed HP-YOLOv8 model in remote sensing images. As depicted in Table 3, HP-YOLOv8 outperforms YOLOv8 in all tested categories. Particularly in the overpass category, HP-YOLOv8 increases the AP from 68.87 to 87.46, an improvement of 18.59 percentage points, demonstrating its high efficiency in handling small and structurally complex objects. Additionally, HP-YOLOv8 raises the AP to 95.82 in the aircraft category, to 98.25 in the oil tank category, and to 98.93 in the playground category, further showcasing the model’s significant advantages in detecting small-sized and hard-to-recognize objects.

Furthermore, to assess the HP-YOLOv8 model’s robustness, we performed experiments on the NWPU VHR-10 and VisDrone2019 datasets, as detailed in Table 4 and Table 5. The experimental results indicate that, in the WPU VHR-10 dataset, HP-YOLOv8 significantly outperforms the traditional YOLOv8 model in the detection of most object categories. Particularly in the ground track field, airplane, basketball court, and vehicle categories, HP-YOLOv8 demonstrates significant performance improvements. For example, in the ground track field category, HP-YOLOv8 increases the AP from 64.73 to 95.45, and in the airplane category, the AP is raised from 92.54 to 99.33, nearly achieving perfect detection results. Additionally, even in generally moderate-performing categories such as basketball court and vehicle, there are notable improvements, with APs rising from 85.28 to 91.84 and from 67.99 to 88.63, respectively. Although there is a slight performance decline in the ship and harbor categories, HP-YOLOv8 overall still demonstrates superior detection capabilities and adaptability.

In the VisDrone2019 dataset, HP-YOLOv8 outperforms YOLOv8 in all categories, particularly excelling in categories such as car, pedestrian, and motor. For instance, the AP for the car category increases from 76.89 to 90.05, for the pedestrian category from 41.37 to 60.30, and for the motor category from 44.82 to 60.41. Additionally, HP-YOLOv8 also performs exceptionally well in typically challenging categories like tricycles and awning tricycles, with AP improvements from 11.35 to 37.55 in the bicycle category, and from 14.10 to 30.27 in the awning tricycles category.

Overall, HP-YOLOv8 generally surpasses YOLOv8 on most performance metrics, particularly demonstrating significant advantages in handling complex scenes and small objects. Concurrently, HP-YOLOv8 proves its exceptional detection capabilities and adaptability, especially in identifying complex or small-sized objects. These test results not only highlight the efficiency of the HP-YOLOv8 model but also confirm its robustness in practical applications.

Convergence Analysis. To assess how well the HP-YOLOv8 model converges on datasets, comprehensive comparisons were conducted between YOLOv8 and HP-YOLOv8 using the RSOD training and validation sets. The investigation centered on evaluating convergence trends for four critical performance metrics: recall, precision, mAP@0.5, and mAP@[0.5:0.95].

As shown in Figure 8, approximately 15 epochs after training begins, HP-YOLOv8 outperforms YOLOv8 across all metrics and stabilizes around 50 epochs. These results clearly demonstrate that HP-YOLOv8 demonstrates superior convergence performance over the traditional YOLOv8 model, providing more stable and reliable performance enhancements. Additionally, the precision–recall (PR) curves offer a direct comparison of model performance, as depicted Figure 9.

### 5.3. Ablation Experiment

We conducted multiple training sessions and tests using the RSOD dataset to assess how three optimization strategies—namely, the C2f-DM module, BGFPN feature fusion method, and SMPDIoU optimization loss function—affected the performance of the YOLOv8 baseline model. The experimental findings present the performance results of different combinations of modules, detailed in Table 6.

In our study, the incorporation of the C2f-DM into the backbone network leads to improvements in the model’s mAP@0.5 from its initial 89.82% to 91.52%, and the mAP@0.5:0.95 also rises from 57.01% to 64.23%. This enhancement validates the efficacy of the C2f-DM module in combining global and local information to enhance the detection of small objects. Substituting the original PANet with BGFPN raises the mAP@0.5 to 92.56% and mAP@0.5:0.95 to 67.78%, while reducing the model’s parameter count by 43.31%. This change demonstrates that BGFPN, with its efficient hierarchical aggregation of network connections, not only significantly boosts mAP performance but also effectively reduces the model’s parameter size. Introducing the SMPDIoU optimization loss function increases the mAP@0.5 to 91.45% and mAP@0.5:0.95 to 64.12%. When combining the C2f-DM module and BGFPN, performance further improves, with mAP@0.5 rising to 93.98% and mAP@0.5:0.95 to 69.78%. By employing all three techniques together, the model achieves its highest performance, with mAP@0.5 reaching 95.11% and mAP@0.5:0.95 reaching 72.03%. These findings demonstrate that HP-YOLOv8 effectively enhances the original YOLOv8 performance. HP-YOLOv8, in contrast to the original YOLOv8, is lighter and more suitable for deployment on hardware-constrained devices.

### 5.4. Comparison with Other Models

To validate the advantages of our proposed HP-YOLOv8 model, we conducted a series of comparative experiments on the RSOD dataset to assess its performance. These experiments not only included comparisons with the latest versions of the YOLO series (YOLOv9 [57] and YOLOv10 [58]) but also involved comparisons with YOLOv8 and several classical detection algorithms (including Dynamic R-CNN [59], CenterNet [32], Fast R-CNN [60], and Cascade R-CNN [31]), specifically designed for remote sensing object detection (such as LSKNet [37], SuperYOLO [39], TPH-YOLO [38], and LAR-YOLOv8 [25]). All models were trained and tested under the same conditions, and the comparative outcomes are displayed in Table 7; the evaluation criteria encompassed metrics such as mAP@0.5, mAP@0.5:0.95, F1 score, parameter count, and Frames Per Second (FPS).

Although the FPS of HP-YOLOv8 is slightly lower than that of the original YOLOv8, TPH-YOLO, and the latest YOLOv9 and YOLOv10, it still surpasses other algorithms in speed. HP-YOLOv8 has a parameter count of 28.52M, which, while higher than YOLOv10, is lower than other models with a higher mAP, such as YOLOv8, LAR-YOLOv8, and SuperYOLO. This indicates that our model achieves high-precision detection at a lower computational cost. Importantly, HP-YOLOv8 demonstrates notable advantages in mAP@0.5 and mAP@0.5:0.95, as well as in parameter efficiency. Specifically, in mAP@0.5:0.95, HP-YOLOv8 records a significant increase of 17.58% compared to the least effective model, Faster R-CNN, conclusively establishing our model’s superior performance and practical value.

### 5.5. Experimental Results Presentation

Ultimately, to visually highlight HP-YOLOv8’s improved detection capabilities, we displayed the detection outcomes of YOLOv8 and HP-YOLOv8 on the RSOD, NWPU VHR-10, and VisDrone2019 datasets. Reviewing Figure 10, we observe the following points: First, the comparison between Figure 10a(1) and Figure 10A(1) reveals that, due to the significant resemblance between the object and the background, the YOLOv8 model fails to detect the aircraft. However, the HP-YOLOv8 model demonstrates its enhanced detection capabilities by not only successfully identifying the object but also clearly distinguishing the background from the foreground, showcasing HP-YOLOv8’s significant advantage in handling background noise. Second, the contrast between Figure 10c(4) and Figure 10C(4) reveals that, due to the partial occlusion of some objects, YOLOv8 exhibits certain missed and false detections. However, HP-YOLOv8 effectively resolves this issue, accurately identifying primary objects and detecting small objects partially obscured by crowds, cars, and trees, especially in cases of significant overlap between objects. These experimental results convincingly prove the significant effectiveness of our proposed HP-YOLOv8 model in enhancing the precision of remote sensing object extraction.

## 6. Conclusions

This paper presented a small-size object detection algorithm for remote sensing images, which builds on the existing YOLOv8 framework by incorporating the newly proposed C2f-DM module, BGFPN feature fusion technology, and SMPDIoU loss function. Through these innovations, we developed HP-YOLOv8 and addressed issues present in YOLOv8 and other current small-size object detection algorithms. Additionally, we conducted extensive testing and comparisons on the RSOD, NWPU VHR-10, and VisDrone2019 datasets. Analysis and experimental validation confirm the effectiveness of each optimized component. HP-YOLOv8 outperformed other detectors in both precision and processing speed, particularly in capturing small objects across various complex scenarios, significantly enhancing the model’s mAP Moving forward, we plan to continue our in-depth research on target detection technologies, striving to surpass existing detectors in precision across all object sizes.

## Figures and Tables

**Figure 1 sensors-24-04858-f001:**
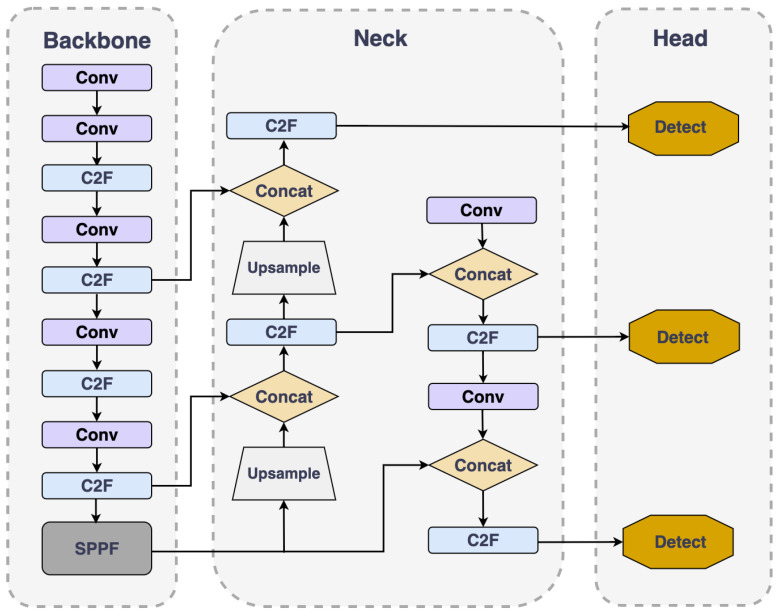
Structure of YOLOv8.

**Figure 2 sensors-24-04858-f002:**
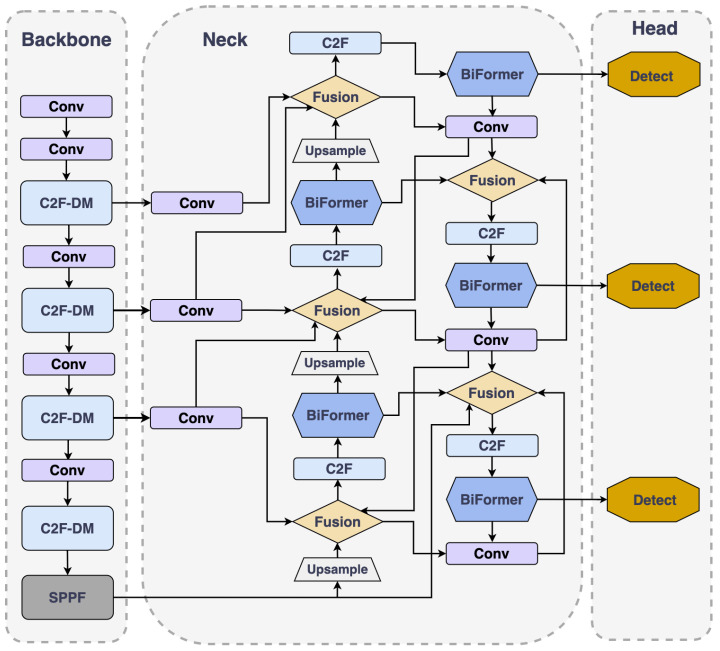
Structure of HP-YOLOv8.

**Figure 3 sensors-24-04858-f003:**
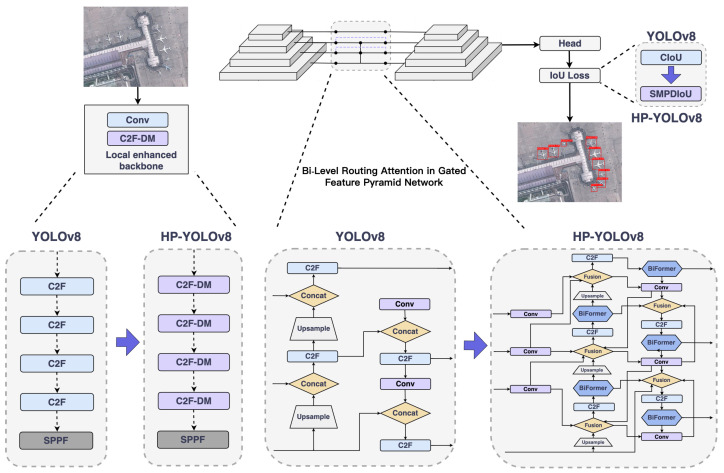
Comparison of YOLOv8 and HP-YOLOv8. C2f-DM (detailed in Section 4.2), BGFPN (detailed in Section 4.3), and SMPDIoU (detailed in Section 4.4).

**Figure 4 sensors-24-04858-f004:**
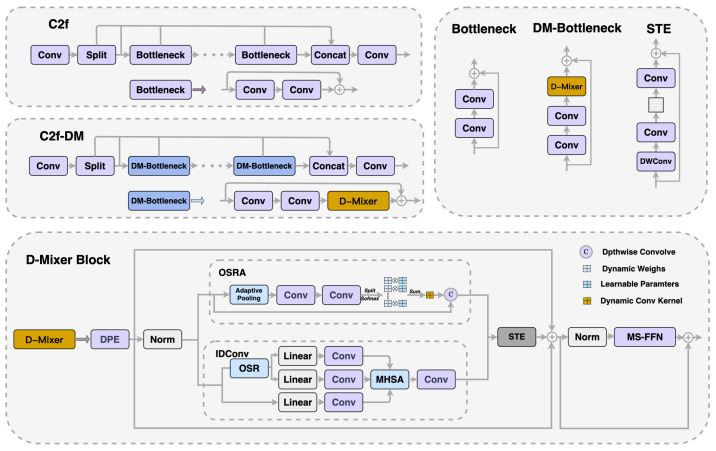
Comparison of C2f and C2f-DM Structure.

**Figure 5 sensors-24-04858-f005:**
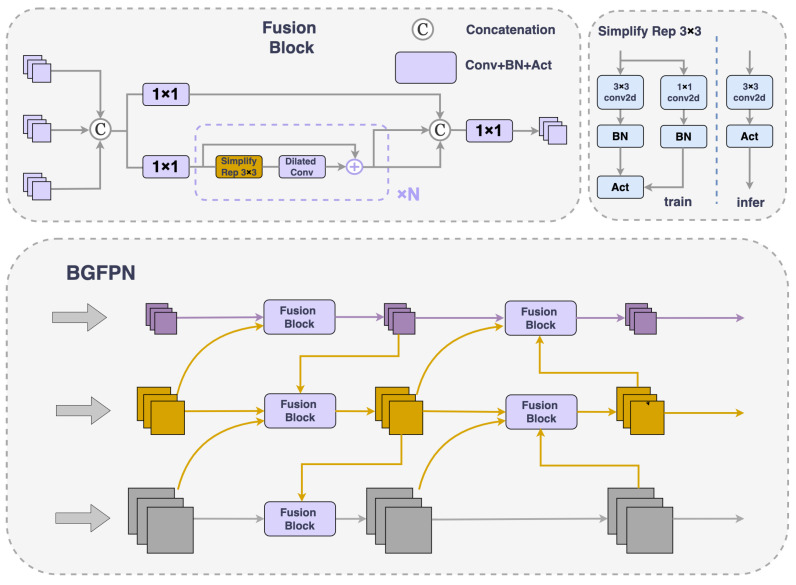
Bi-Level Routing Attention in Gated Feature Pyramid Network.

**Figure 6 sensors-24-04858-f006:**
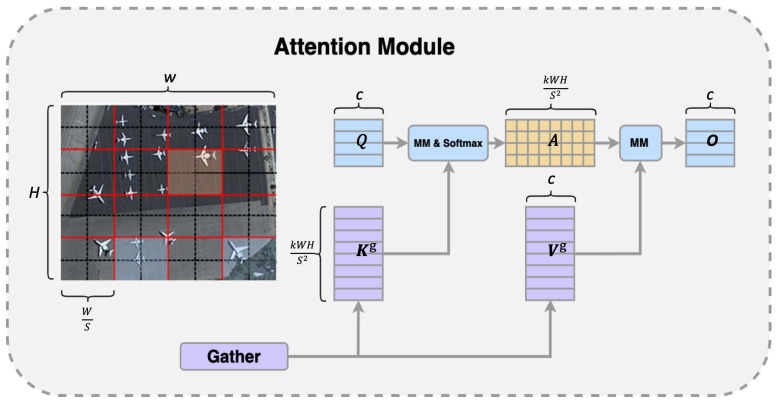
Bi-Level Routing Attention.

**Figure 7 sensors-24-04858-f007:**
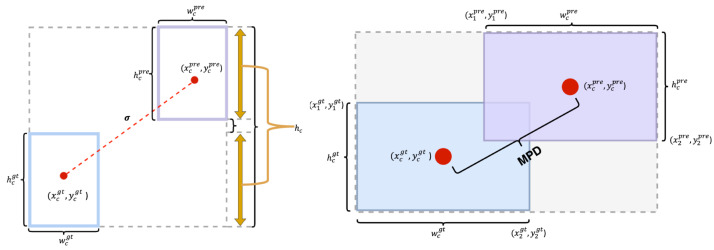
Distance loss diagram and MPD schematic diagram.

**Figure 8 sensors-24-04858-f008:**
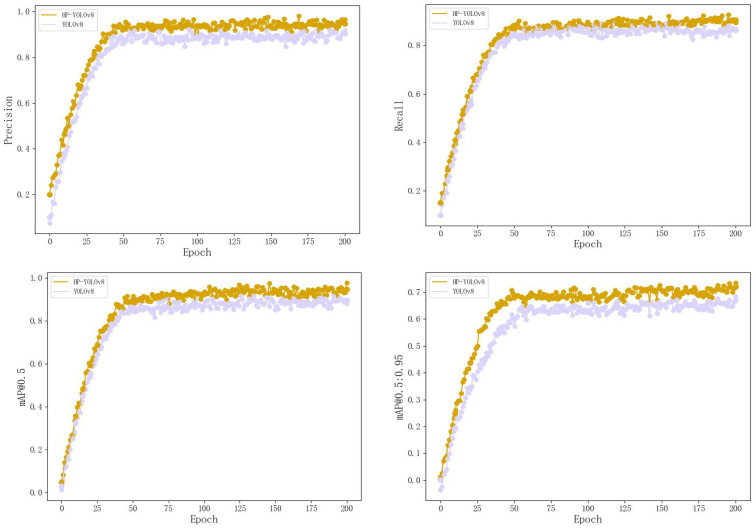
Evaluation of trends in recall, precision, mAP@0.5, and mAP@[0.5:0.95] for YOLOv8 and HP-YOLOv8 on the RSOD validation dataset.

**Figure 9 sensors-24-04858-f009:**
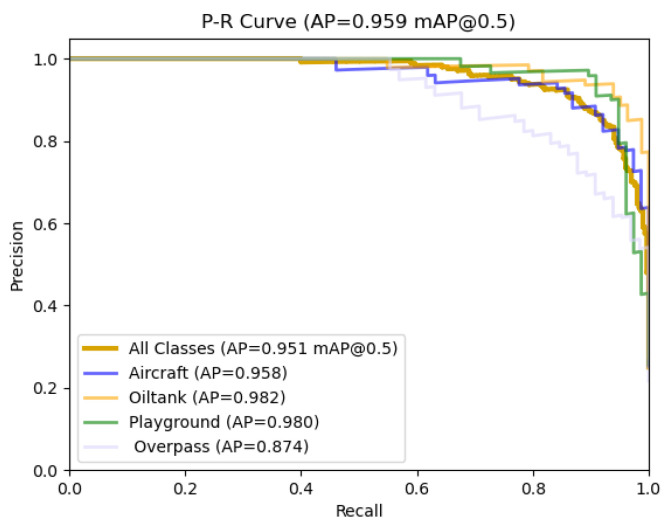
Precision–recall curves for RSOD datasets.

**Figure 10 sensors-24-04858-f010:**
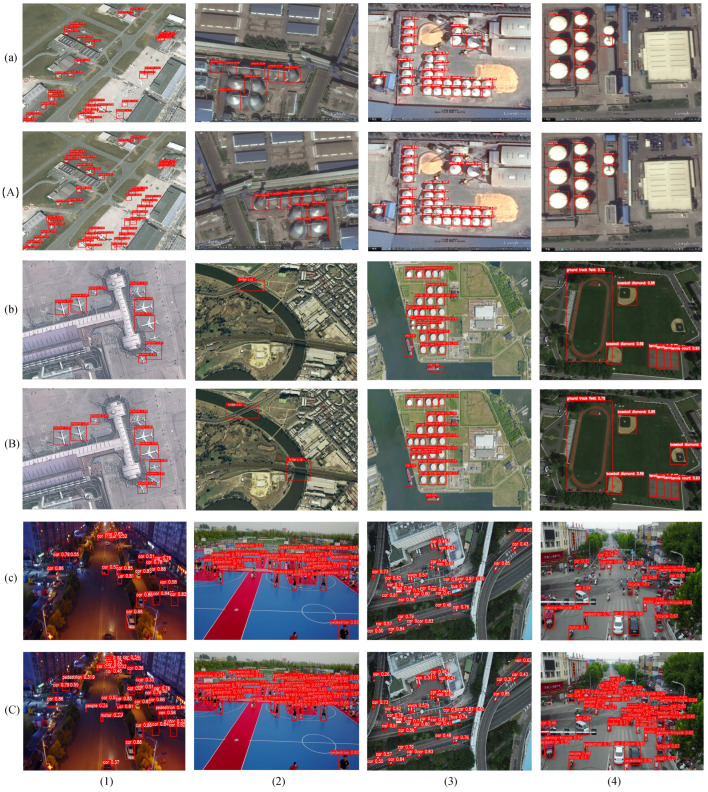
Detection outcomes on the RSOD, NWPU VHR-10, and VisDrone2019 datasets are depicted. Panels (**a**–**c**) show the results using YOLOv8, whereas panels (**A**–**C**) illustrate the results from HP-YOLOv8.

**Table 1 sensors-24-04858-t001:** Experimental environment configuration table.

Configuration Item	Name	Specification
Hardware environment	GPU	NVIDIA GeForce RTX 3080
	CPU	Intel Core i7-11700K
	VRAM	12G
	RAM	64G
Software environment	Operating System	Ubuntu 18.04
	Python	3.8.12
	Pytorch	1.10.0
	CUDA	10.4
	cuDNN	7.6.5

**Table 2 sensors-24-04858-t002:** Model training hyperparameter settings.

Hyperparameter Options	Setting
Epochs	200
Initial Learning Rate 0	0.01
Learning Rate Float	0.01
Input Resolution	640 × 640 × 3
Weight_decay	0.0005
Momentum	0.937
Batch_size	4

**Table 3 sensors-24-04858-t003:** Peformance of HP-YOLOv8 and YOLOv8 on RSOD.

Model	Class	Aircraft	Oil tank	Overpass	Playground
YOLOv8	P	95.52	96.83	71.92	95.31
	R	91.62	95.34	70.21	96.82
	F1	93.53	96.06	71.06	96.06
	AP	95.34	97.05	68.87	98.02
HP-YOLOv8	P	97.23	96.85	87.42	96.65
	R	90.76	95.23	81.94	97.23
	F1	93.93	96.62	84.62	96.94
	AP	95.82	98.25	87.46	98.93

**Table 4 sensors-24-04858-t004:** Peformance of HP-YOLOv8 and YOLOv8 on NWPU VHR-10.

Model	Class	Bridge	Ground Track Field	Ship	Baseball Diamond	Airplane	Basketball Court	Vehicle	Tennis Court	Harbor	Storage Tank
YOLOv8	P	95.95	76.84	98.65	93.89	94.56	89.94	90.12	93.21	98.45	92.82
	R	80.23	54,76	94.78	92.56	85.80	70.64	64.87	85.46	99.25	82.98
	F1	87.47	63.93	96.68	93.22	90.04	79.16	75.93	89.17	98.85	87.70
	AP	90.73	64.73	99.01	95.30	92.54	85.28	67.99	91.84	96.17	86.56
HP-YOLOv8	P	96.87	97.56	98.4	92.34	96.45	94.87	91.96	95.45	98.78	93.71
	R	86.65	97.50	93.97	93.48	97.89	87.62	73.45	87.21	98.86	80.67
	F1	91.53	97.53	96.15	92.91	97.17	91.16	81.81	91.08	98.16	86.77
	AP	91.15	95.45	98.32	96.66	99.33	91.84	88.63	92.06	95.84	89.20

**Table 5 sensors-24-04858-t005:** Peformance of HP-YOLOv8 and YOLOv8 on VisDrone2019.

Model	Class	Van	Pedestrian	Car	Bicycle	Person	Motor	Bus	Tricycle	Truck	Awning Tricycle
YOLOv8	P	48.47	46.87	84.98	13.78	38.07	50.26	61.72	31.88	42.87	17.87
	R	38.74	35.89	71.28	8.32	26.81	41.63	52.42	23.69	30.76	10.43
	F1	43.29	40.90	77.73	10.64	31.67	45.54	56.74	27.38	36.12	13.32
	AP	42.75	41.37	76.89	11.35	29.78	44.82	56.32	26.93	35.49	14.10
HP-YOLOv8	P	62.86	63.56	92.43	42.65	53.78	63.41	73.90	44.98	47.88	37.65
	R	52.56	58.72	90.02	35.62	44.69	58.32	67.54	34.12	41.42	28.64
	F1	57.35	61.05	91.21	38.90	48.95	60.80	70.59	39.05	44.48	32.64
	AP	57.45	60.30	90.05	37.55	48.22	60.41	69.77	37.62	43.33	30.27

**Table 6 sensors-24-04858-t006:** Ablation experiment of different components in HP-YOLOv8.

Model				Params	FPS	P	R	F1	mAP@0.5	mAP@0.5:0.95
YOLOv8	C2f-DM	BGFPN	SMPDIoU							
🗸				43.41 M	75.78	89.18	89.27	89.22	89.82	57.01
🗸	🗸			44.14 M	63.35	89.86	91.36	90.60	91.52	64.23
🗸		🗸		24.61 M	60.49	91.78	92.41	92.09	92.56	67.78
🗸			🗸	43.61 M	75.78	90.05	91.54	90.79	91.45	64.12
🗸	🗸	🗸		28.52 M	55.46	91.89	93.78	92.82	93.98	69.78
🗸	🗸	🗸	🗸	28.52 M	55.46	92.21	94.22	93.21	95.11	72.03

**Table 7 sensors-24-04858-t007:** Comparison with different models.

Model	P	R	F1	mAP@0.5	mAP@0.5:0.95	Params	FPS
Faster R-CNN [60]	87.78	82.39	84.97	85.46	54.45	42.47 M	31.73
Cascade R-CNN [31]	89.54	84.0	86.73	86.21	55.31	70.62 M	26.48
CenterNet [32]	87.92	86.54	87.23	87.79	56.14	33.34 M	34.37
Dynamic-RCNN [59]	87.36	82.88	85.07	85.30	55.86	42.78 M	31.35
LSKNet [37]	88.05	85.37	86.70	87.74	56.35	29.88 M	48.75
TPH-YOLO [38]	90.52	89.79	90.15	90.46	57.32	53.59 M	56.26
SuperYOLO [39]	91.89	90.21	91.05	90.78	59.30	54.66 M	32.21
LAR-YOLOv8 [25]	94.04	90.92	92.46	92.92	61.55	28.56 M	54.89
YOLOv8	91.38	86.32	88.80	89.82	57.76	44.60 M	75.78
YOLOv9 [57]	93.45	89.65	91.53	92.05	60.48	38.78 M	79.64
YOLOv10 [58]	94.47	91.02	92.73	93.35	65.82	28.21 M	85.71
HP-YOLOv8 (Ours)	96.75	93.05	94.88	95.11	72.03	28.52 M	55.46

## Data Availability

Data are contained within the article.

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
