# Peer review of "HP-YOLOv8: High-Precision Small Object Detection Algorithm for Remote Sensing Images"

_sensors, 2024, doi:10.3390/s24154858_

Round 1

Reviewer 1 Report

Comments and Suggestions for Authors

The manuscript introduces the C2f-DM module as a replacement for the original C2f module, claiming it integrates local and global information better.

The BGFPN and BRA mechanisms are proposed as enhancements, but the manuscript does not explain in detail how these techniques uniquely address the challenges of small object detection in remote sensing images compared to existing methods.

Experiments are conducted on RSOD, NWPU VHR-10, and VisDrone2019 datasets. The manuscript does not discuss whether these datasets comprehensively represent the variety of scenarios in remote sensing images, which may limit the generalizability of the results.

The manuscript primarily uses [email protected] and [email protected]:0.95 as evaluation metrics. Including additional metrics such as F1-score, precision, and recall would provide a more holistic evaluation of the model's performance.

Reviewer 2 Report

Comments and Suggestions for Authors

In this paper,the authors present HP-YOLOv8, which builds on YOLOv8,to identify small objects swiftly and precisely.C2f-D-Mixer module integrates both local and global information,improving the ability to detect features of small objects.And the Bi-Level Routing Attention in Gated Feature Pyramid Network effectively captures critical feature information of small target objects.Finally,they propose a new loss function,which provide a more accurate bounding box regression loss calculation method.Overall, the article is innovative and its presentation is good. However, there are still some problems to be improved:

1.Figure 1 and Figure 2 should be combined for comparison to better illustrate the differences and connections between the two models.

2.In the main text, abbreviations should be explained in detail when they first appear, such as GAM(in 2.2) and BRA(in introduction).

3.Figure 3 is a bit messy. Please revise it to clearly show the relationships between DM-Block and HP-YOLOv8, as well as between STE and DM-Block.Additionally, you should explain the differences between DM-Bottleneck and Bottleneck.

 4.In Figure 5, you introduced the term "attention moddul"(which should be attention attention module). This is a very basic mistake. Please carefully review the entire text to avoid such typos.

 5.The paper discusses the detection of small objects, but the absolute and relative small objects do not match the small objects mentioned in your text. For instance, the ground track field detected in Figure 9 is difficult to associate with small objects. Please provide a detailed and convincing explanation.

 6.This paper builds a new model based on YOLOv8. However, YOLOv9 was released six months ago, and YOLOv10 has been available for a long time. This paper could include comparative experiments with these versions.

7. Some recent works should be discussed, including "Multiple information perception-based attention in YOLO for underwater object detection", "Manifold-based Incomplete Multi-view Clustering via Bi-Consistency Guidance" and "Graph-Collaborated Auto-Encoder Hashing for Multi-view Binary Clustering".

8.Figure 9 sequentially presents the model's detection outcomes from three datasets. In c and C (VisDrone2019), a great deal of vehicles are detected, but VisDrone2019 does not include  vehicle category. Additionally,In c (1) and (2), there is an issue where each instance of a person and vehicle combined together is detected as a new vehicle. Please carefully verify this and provide a detailed and convincing explanation.

9.In Figure 9, b(2) and B(2) detected airplanes with high confidence, but I did not see any airplane,only bridges. This is a serious issue.Please carefully verify this and provide a detailed and convincing explanation.

Reviewer 3 Report

Comments and Suggestions for Authors

Review Report: HP-YOLOv8: High-Precision Small Object Detection Algorithm for Remote Sensing Images

The manuscript presents an improved object detection algorithm HP-YOLOv8 for detecting small objects in remote sensing images. The proposed modifications to the YOLOv8 framework are demonstrated to improve performance across multiple datasets. The experimental results are comprehensive and indicate improvements over the baseline YOLOv8 model. The manuscript is well-organized and clearly presents the methodology, experiments, and results. I did not find any flaws in the analysis. I recommend accepting the paper as it is.

Reviewer 4 Report

Comments and Suggestions for Authors

This work proposes an enhanced algorithm optimized for detecting small objects in remote sensing images. Many interesting results are presented and compared with those predicted by other methods in this work. It seems the proposed method have several merits over others. This work is valuable for many applications in this field and appropriate to the Journal Sensors. I recommend its publication on Sensors in its current form.

Comments on the Quality of English Language

The language is fine.

Round 2

Reviewer 1 Report

Comments and Suggestions for Authors

The revised version of the manuscript can be acceptable.